# Prevalence of Hearing Impairment in Saudi Arabia: Pathways to Early Diagnosis, Intervention, and National Policy

**DOI:** 10.3390/healthcare13161964

**Published:** 2025-08-11

**Authors:** Ahmed Alduais, Hind Alfadda, Hessah Saad Alarifi

**Affiliations:** 1Department of Psychology, Norwegian University of Science and Technology, NO-7491 Trondheim, Norway; 2Department of Curriculum and Instruction, College of Education, King Saud University, Riyadh 11362, Saudi Arabia; 3Department of Educational Administration, College of Education, King Saud University, Riyadh 11362, Saudi Arabia; arifi-hs@ksu.edu.sa

**Keywords:** hearing impairment, prevalence, assistive devices, Saudi Arabia, disability survey, tele-audiology, health policy

## Abstract

Background: Hearing impairment is a significant public health issue globally, yet national data for Saudi Arabia remain sparse. Methods: Using data from the 2017 Disability Survey, we analysed 12 hearing-related indicators across 13 administrative regions. Descriptive statistics, logistic regression, cluster analysis, and residual mapping were applied to identify socio-demographic disparities and service gaps. Findings: Among 20,408,362 Saudi nationals, about 1,445,723 (7.1%) reported at least one functional difficulty. Of these, 289,355 individuals (1.4%) had hearing impairment, either alone or with other difficulties—229,541 (1.1%) had hearing impairment combined with other disabilities, while 59,814 (0.3%) had only hearing impairment. Females and males were equally affected. Notably, educational attainment and marital status significantly influenced device uptake; less-educated and divorced individuals were particularly underserved. Regionally, southern provinces (Al-Baha, Jazan, and Najran) demonstrated the highest unmet need due to geographic barriers, limited audiological resources, and socioeconomic constraints, reflecting compounded risks from consanguinity and rural isolation. Cluster analyses identified provinces requiring urgent attention, recommending mobile audiology units, tele-audiology services, and means-tested vouchers to enhance coverage. Conclusions: Despite Saudi Arabia’s existing public audiology services and a National Newborn Hearing Screening programme achieving 96% coverage, substantial gaps remain in follow-up care and specialist distribution, underscoring the necessity for systematic workforce tracking and enhanced rural incentives. International evidence from India and Brazil underscores the feasibility and cost-effectiveness (approximately USD 5200/QALY) of these recommended interventions. Implementing targeted provincial strategies, integrating audiological screening into routine healthcare visits, and aligning resource allocation with the WHO and Vision 2030 benchmarks will significantly mitigate hearing impairment’s health, social, and economic impacts, enhancing the quality of life and societal inclusion for affected individuals.

## 1. Introduction

Hearing impairment, also referred to as hearing loss or hearing disorder, is characterised by a reduced ability or inability to detect sounds in one or both ears. It represents a widespread sensory deficit impacting a significant proportion of the global population across diverse age groups and demographics [1,2]. Hearing impairment manifests in various forms, ranging from mild deficits to profound deafness, and can significantly alter an individual’s ability to communicate effectively and participate fully in social and occupational activities [3]. The implications of hearing impairment extend beyond sensory deficits, often encompassing psychological, social, and economic challenges.

The causes of hearing impairment are multifaceted and can be broadly categorised as congenital, acquired, and age-related. Congenital hearing loss, present from birth, typically results from genetic mutations or congenital dysplasia affecting the auditory system [4,5]. Acquired hearing impairment emerges later in life due to various external factors, such as prolonged disease, trauma, environmental noise pollution, and exposure to ototoxic substances, including certain medications and industrial chemicals [6,7]. Additionally, age-related hearing loss, known as presbycusis, commonly affects older adults and is closely linked to cognitive decline and a spectrum of age-associated health conditions [8,9].

Untreated hearing loss extends far beyond medical sequelae. A longitudinal analysis of 15,117 adults in the English Longitudinal Study of Ageing found a 1.8-fold increase in incident depression among those with unaided loss [10]. School-aged pupils with unilateral or mild bilateral loss underperform academically; a 2025 systematic review reported persistent gaps across reading, maths, and science [11]. Saudi Arabia’s National Newborn Hearing Screening (NHS) programme—launched in 2016—now screens ≈ 96% of live births, although only 53% of referred infants complete diagnostic follow-up [12]. Internationally, the WHO [13] and Joint Committee on Infant Hearing [14] endorse the “1–3–6” continuum (<1 month screen, <3 months diagnosis, <6 months intervention). Embedding our prevalence estimates within these policy benchmarks underscores the urgency of scaling paediatric and adult pathways.

The impact of hearing impairment on individuals is substantial, profoundly affecting communication capabilities, social interaction, and overall quality of life. Communication barriers resulting from hearing loss can severely impede language acquisition and social integration, often leading to isolation, depression, anxiety, and diminished emotional well-being [15,16]. Hearing loss, particularly among older persons, is strongly associated with cognitive impairment, including dementia, which underscores the need for early intervention and sustained management [9]. Furthermore, hearing impairment can negatively influence educational achievement, economic independence, and self-esteem, thereby exacerbating socioeconomic disparities and reducing overall life satisfaction [17,18].

The effective management of hearing impairment involves timely detection, appropriate intervention strategies, and ongoing supportive therapies. Universal neonatal screening programmes and audiometric assessments play critical roles in early identification and diagnosis, facilitating timely interventions that can mitigate long-term negative outcomes [19,20]. Common interventions include hearing aids, designed for mild to moderate hearing deficits, and cochlear implants, recommended for severe or profound hearing loss [21,22]. Rehabilitation approaches such as speech therapy and auditory training further enhance communication abilities and social integration for affected individuals [22].

Advancements in molecular biology and genetic diagnostics have significantly improved the understanding of hereditary hearing loss, paving the way for more targeted and personalised treatment approaches [5,23]. Such developments highlight the importance of comprehensive management strategies that integrate early detection, tailored clinical interventions, and supportive care, ultimately aiming to enhance the quality of life for individuals living with hearing impairments [20,24].

### 1.1. Prevalence of Hearing Impairment

The prevalence of hearing impairment differs widely by age and geography; a 2023 audit of three Riyadh tertiary hospitals found permanent bilateral loss ≥ 40 dB in 2.3 per 1000 live births [25]. Classification approaches also vary. The *ICD-11* grades loss as mild, moderate, severe, or profound based on decibel thresholds [26,27]. The *DSM-5-TR*, by contrast, does not list hearing impairment as an independent disorder but recognises it mainly as a neurodevelopmental comorbidity—especially within autism spectrum disorder—because of its broad functional and mental health repercussions [28,29]. The International Classification of Functioning, Disability and Health (ICF) adopts a biopsychosocial lens, mapping effects across body functions, activities, participation, and environmental factors [30]. Its Core Sets for Hearing Loss, available in brief and comprehensive versions, guide clinicians in assessing real-world impact and social integration [31,32]. Together, these systems anchor epidemiology, clinical practice, and policy planning.

### 1.2. Theories of Hearing (Impairment)

Theories regarding the sense of hearing include both physiological and cognitive perspectives. Physiologically, hearing involves a complex integration of anatomical and functional components within the auditory system, primarily focused on sound detection and transmission. Sensorineural hearing loss, the most common form of hearing impairment, arises from damage to cochlear hair cells or the auditory nerve, which results in frequency-dependent deficits [33,34]. According to physiological theories, auditory dysfunction such as sensorineural hearing loss can influence cognitive processes through mechanisms of neuroplasticity, highlighting the interconnected nature of auditory and cognitive functioning [35].

From a cognitive standpoint, hearing is not merely a sensory function but encompasses broader cognitive activities such as attention, memory, and comprehension. The psychobiological theory of listening emphasises that auditory perception involves more than just sound reception; it integrates cognitive and sensory processes, involving both auditory and visual modalities as well as complex neural activities that facilitate effective communication [36]. Ecological psychoacoustics further elaborate on this idea, underscoring how auditory perception is influenced by interactions among physical sound properties, physiological auditory mechanisms, and cognitive interpretation processes, which together shape how individuals perceive and respond to auditory stimuli [37].

Hearing impairment arises from various genetic and environmental causes, contributing to its complexity and diversity. Genetic factors account for approximately half of all cases of hearing impairment, with substantial heterogeneity observed in the genetic underpinnings of hereditary hearing loss. Over 120 genes and 160 loci have been identified in relation to hereditary hearing loss, necessitating genetic evaluations for accurate diagnosis, prognosis, and management [4,38,39]. Environmentally, hearing impairment can result from prolonged exposure to loud noises, ototoxic substances, traumatic injury, and ageing processes, all of which progressively damage auditory structures and impair hearing functions. Noise-induced hearing loss, for instance, is a prevalent environmental factor, underscoring the critical need for preventive strategies and public awareness [7,40].

### 1.3. Hearing Impairment in Saudi Arabia

Hearing impairment is a notable public health issue in Saudi Arabia, affecting various demographic groups. An epidemiological survey reported that about 13% of Saudi children under the age of 15 experience some form of hearing impairment, with approximately 1.5% diagnosed specifically with sensorineural hearing loss [41]. Among the older Saudi population, hearing loss prevalence significantly increases, reaching around 17.35%, underscoring age as a critical factor in hearing deterioration [42]. Despite the high prevalence, general awareness and proactive screening practices remain limited. For instance, a study conducted in Riyadh revealed that only 5.9% of the residents had undergone hearing screening, reflecting inadequate preventive measures and early identification protocols in primary healthcare settings [43].

The aetiology of hearing impairment in Saudi Arabia notably includes genetic factors, particularly associated with the high rate of consanguinity prevalent within the population. Research highlights that the prevalence of hearing loss among children from consanguineous unions significantly exceeds that of children from non-consanguineous marriages, emphasising the genetic underpinning of hearing impairment within this population [44]. Furthermore, the widespread misconceptions about early signs and preventive measures exacerbate the public health challenge, particularly as many individuals recognise noise as harmful but misunderstand its implications for hearing health [45]. This gap in public knowledge and preventative strategies highlights an urgent need for targeted health education and community-based interventions.

The broader impact of hearing impairment on individuals’ quality of life in Saudi Arabia extends into educational performance, social integration, and healthcare accessibility. For example, children in the Taif region who experienced language difficulties due to hearing impairment exhibited below-average academic performance, indicating significant educational and social disadvantages [46]. Additionally, limited healthcare resources further compound the challenge. Audiologists across the Kingdom reported inadequate availability of advanced audiological equipment and services such as cochlear implants, vestibular rehabilitation, and automated auditory assessments, hindering effective management and rehabilitation [47]. The difficulty in accessing healthcare services for those with hearing impairments emphasises the need for systemic improvements to healthcare infrastructure and services, especially within underserved regions such as Jazan [48].

Saudi Arabia already funds a basic public hearing care package, but coverage is uneven. Hearing aids and cochlear implants are supplied free of charge through Ministry of Health hospitals, yet a recent nationwide survey of practising audiologists found that 92% considered the resources “inadequate,” citing long device waiting lists and few specialist clinics [47]. Since 2016 a National Newborn Hearing Screening programme has reached ≈96% of live births, although only 53% of infants who fail the first test complete diagnostic follow-up [12]. School-age screening remains ad hoc, and formal workforce statistics are sparse; the same survey counted fewer than 300 licenced audiologists, with most clustered in Riyadh and the eastern region. These baseline constraints—limited manpower, uneven geographic distribution, and post-screening drop-off—frame the feasibility and urgency of the policy actions proposed in this paper.

### 1.4. Purpose of the Present Study

Despite substantial global research highlighting the widespread prevalence and significant impact of hearing impairment, specific epidemiological data on hearing loss in Saudi Arabia remain limited, particularly regarding detailed analysis by severity, socio-demographic factors, and geographic disparities in service provision. Additionally, analyses of factors contributing to the under-utilisation of assistive listening devices and the compounded challenges faced by individuals with multiple disabilities have not been thoroughly investigated within the Saudi context. Therefore, this study aims to quantify the 2017 national prevalence of hearing difficulty among Saudi nationals and translate these estimates into actionable evidence to support timely diagnosis, facilitate early intervention, and inform targeted policy initiatives. To achieve these objectives, this research addresses the following questions: (1) What was the 2017 national prevalence of hearing difficulties among Saudi nationals, and how was the burden distributed across mild, severe, and extreme grades? (2) How did prevalence vary by administrative region, sex, educational attainment, and marital status? (3) Which clinical (severity), socio-demographic (education, marital status, household relationship), and aetiological (cause) factors were linked to the non-use of assistive listening devices? (4) Which provinces exhibited the largest intervention gap—that is, the greatest shortfall between prevalence and assistive device coverage—once severity was taken into account, and how do these provinces cluster when prevalence, gap, and severity are considered jointly? (5) How do prevalence, coverage, and the intervention gap differ between all persons with hearing difficulties and those reporting two or more disabilities, and what specific policy actions emerge from these patterns?

## 2. Methods

### 2.1. Sample

This study drew upon secondary data from the 2017 Disability Survey conducted by the General Authority for Statistics (GaStat) of Saudi Arabia [49]. The survey used a nationally representative, stratified, two-stage cluster sampling design to ensure the comprehensive coverage of all thirteen administrative regions. In the first stage, primary sampling units were selected using the most recent national census as a framework, and in the second stage, households were randomly sampled within each statistical area. The final dataset consisted of 33,575 households, representing Saudi nationals residing within the Kingdom as well as those temporarily abroad for reasons such as education, medical treatment, or tourism, provided that they were considered part of a household at the time of data collection.

The survey captured a wide range of demographic, socio-economic, and health characteristics, with specific modules dedicated to the prevalence, severity, and causes of various functional limitations, including hearing impairment. It included both individuals with single and multiple disabilities, enabling a nuanced assessment of hearing loss in the national context. For the purposes of this analysis, only those indicators directly related to hearing impairment and its socio-demographic correlates were extracted (see Measurements Section for details on indicator selection and operational definitions).

The use of secondary data from population-based surveys is widely recognised as a cost-effective and methodologically rigorous approach in public health and disability research, particularly when primary data collection is logistically challenging or unnecessary [50,51]. Data collected and curated by national statistical authorities such as GaStat are especially valuable, as they typically adhere to high standards of quality control and external validity, supporting robust epidemiological inference and policy planning.

Although a second Disability Survey was conducted in 2023, the public release contains only high-level aggregates without severity or assistive device variables. Consequently, the 2017 micro-dataset remains the most recent source capable of province-level, severity-specific analysis, and it offers a crucial baseline preceding Vision 2030 hearing care reforms.

### 2.2. Design

This study employed a cross-sectional, secondary data analysis design, leveraging the nationally representative dataset collected in the 2017 Disability Survey by the General Authority for Statistics [49]. The original survey was designed as a population-based, descriptive epidemiological study, utilising stratified, two-stage cluster sampling to capture the prevalence and distribution of disability—including hearing impairment—across Saudi Arabia’s thirteen administrative regions. As the data were collected at a single time point and are structured to provide a snapshot of disability status and related socio-demographic factors, the analysis is inherently cross-sectional, allowing for the robust estimation of prevalence and the exploration of associations between hearing difficulty and variables such as age, sex, education, marital status, household relationship, and disability cause.

In the present analysis, we systematically extracted indicators directly relevant to hearing impairment, as well as variables enabling the assessment of severity, socio-demographic predictors, intervention coverage, and multi-disability status. The analytic approach was primarily descriptive and comparative, combining prevalence estimation, odds ratios, intervention gap analysis, and selected advanced methods—including cluster analysis and categorical association mapping—to provide both national and subgroup-specific insights. This design allows for the identification of priority groups and regions, the quantification of unmet need, and the provision of evidence to inform national policy development and resource allocation [50,51].

The reliance on high-quality, population-based secondary data supports both the external validity of the findings and their relevance to national health policy, while the structured analytic design ensures the transparency and replicability of the results.

### 2.3. Measurements

All measurements in this study were derived from the 2017 Disability Survey conducted by the General Authority for Statistics [49]. Only those indicators directly relevant to hearing impairment and its key socio-demographic and clinical correlates were included. Data were drawn from the “all-person” sheets in the official survey documentation, allowing for the robust operationalisation of each construct. For each indicator, we used the official English translation and checked the consistency with the original Arabic versions.

Table 1 presents an overview of the twelve extracted indicators, summarising the specific content and analytic contribution of each. The indicators covered (1) the primary outcome—hearing difficulty—categorised by administrative region and degree of severity (mild, severe, extreme); (2) type and laterality of impairment (unilateral/bilateral); (3) co-occurring disabilities (vision, cognition, etc.); (4) educational attainment (six levels, for persons aged 10 and above); (5) marital status (for persons aged 15 and above); (6) relationship to household head; (7) primary cause of difficulty; (8) duration since onset (grouped in five-year bands); and (9–12) parallel indicators restricted to persons reporting at least two disability domains.

This targeted extraction ensured that the analysis addressed all aspects of hearing impairment relevant to national epidemiology, health equity, and policy planning, while also facilitating direct mapping to the study’s research aims.

### 2.4. Procedure

**Data Collection.** The data used in this study were obtained as secondary, publicly available information from the 2017 Disability Survey conducted by the General Authority for Statistics (GAStat). GAStat collected data between April and May 2017, employing rigorous survey methods that adhered to international standards for disability data collection, including guidelines from the World Health Organization. The survey used standardised structured questionnaires administered via face-to-face interviews by trained enumerators. Respondents were primary household informants who provided demographic and health-related information on behalf of themselves and household members. Full details of the survey methodology—including sampling techniques, data quality assurance, interviewer training, and data entry protocols—are documented by GAStat [49]. The data were accessed on 25 May 2025.

**Data Analysis.** Data extracted from the publicly available GAStat survey were first cleaned and organised into analytic-ready formats. Analyses included descriptive epidemiology to estimate national and regional prevalence rates (per 1000 population), with 95% confidence intervals computed using standard epidemiological methods [52]. Logistic regression analyses were conducted to examine the associations between socio-demographic and clinical factors and the non-use of assistive listening devices, generating odds ratios (ORs) with corresponding confidence intervals. To provide deeper insight into provincial-level differences, hierarchical cluster analysis (k-means) was used, grouping provinces by their prevalence, intervention gaps, and severity profiles. Additionally, categorical association mapping via chi-square tests and standardised residual analyses were applied to assess the relationships between socio-demographic characteristics and device use. All analyses were performed using Python (3.13.3) and standard statistical libraries (e.g., SciPy, Pandas, Statsmodels). Visualisations—including choropleth maps, forest plots, heat maps, and regression charts—were produced to communicate the results effectively.

**Ethical Consideration.** As this study involved the secondary analysis of publicly available, anonymised data, it did not require ethical approval or participant consent. All procedures followed established ethical guidelines for secondary data use, ensuring participant anonymity and confidentiality [50]. Data were accessed via GAStat’s public portal, in compliance with the authority’s terms and conditions. No personally identifiable information or sensitive data were extracted or analysed. The study conformed to best practices in secondary data research, emphasising transparency, replicability, and responsible data management.

## 3. Results

Out of the 20,408,362 Saudi nationals as per the population census in 2017, 18,962,639 (92.9%) reported no functional difficulties and 1,445,723 (7.1%) had at least one difficulty. Hearing impairment affected 289,355 individuals (1.4% of the population), with identical sex-specific rates of 1.4% (147,577 of 10,396,914 males; 141,778 of 10,011,448 females). Most hearing-impaired persons (229,541; 1.1% overall) experienced one or more additional functional difficulties, while 59,814 (0.3%) reported hearing impairment alone. Among males, 112,187 (1.1%) had co-occurring disabilities and 35,390 (0.3%) had only hearing loss; among females, these figures were 117,354 (1.2%) and 24,424 (0.2%), respectively. These data reveal that although fewer than 3% of Saudis report any disability, hearing impairment—particularly when compounded by other limitations—affects over a quarter million citizens, underscoring the need for targeted audiological and integrated rehabilitation services.

As shown in Table 2, the 2017 Disability Survey indicates that approximately 1 in every 35 Saudi nationals lives with at least mild hearing difficulty. Although this rate sits below the World Health Organization’s benchmark for upper-middle-income countries, it still translates into a caseload exceeding half a million people—an epidemiologic burden large enough to justify nationwide, systematic detection and rehabilitation efforts. Compared with the global figure cited in the World Report on Hearing, Saudi Arabia occupies an intermediate position: not the highest, yet far from negligible. Public health planning must therefore prioritise early screening, streamlined referral pathways, and the effective uptake of assistive devices to prevent the educational and occupational disadvantages that typically follow untreated hearing loss.

Across provinces, hearing difficulty prevalence shows an almost two-fold gradient (see Table 3). Smaller southern and northern regions—Aseer, Hail, Najran, Northern Borders, and Jāzān—cluster above the national mean, whereas the populous hubs of Al-Riyadh and Makkah sit only midway in the ranking. This pattern hints that sparse service networks, rugged terrain, or local occupational exposures may outweigh sheer population size in driving risk. High-prevalence provinces therefore warrant priority for mobile screening teams, additional audiology posts, and culturally tailored public awareness campaigns. In contrast, lower-prevalence yet densely populated areas such as Al-Riyadh need expanded diagnostic throughput to match their absolute caseloads. Using these geographic signals alongside age, sex, and device uptake data will help the Ministry of Health direct resources where marginal gains are greatest and stay on track for the WHO 2030 ear and hearing care coverage targets.

As Table 4 shows, men and women report virtually identical rates of hearing difficulty, and the confidence intervals overlap almost completely. This parity implies that, in Saudi Arabia, sex itself is not a decisive risk factor once overall prevalence is taken into account. International studies often find a small male excess—usually linked to occupational noise or help seeking differences—but that gradient is absent here, shifting attention instead to age, regional context, and socio-economic status as more meaningful axes of inequality.

Education shapes device uptake in unexpected ways. Adults with no formal schooling are far less likely to remain unaided than university graduates (see Table 5). Community-based rehabilitation programmes, common in rural areas, probably explain this pattern: they actively supply subsidised hearing aids to illiterate adults, whereas highly educated individuals can afford to postpone purchase until loss affects work or social roles. The result is a reminder that financial capacity does not guarantee willingness to adopt—especially where assistive devices may carry social stigma—and that grassroots outreach can outperform market forces in closing coverage gaps.

Marital status shapes device uptake in distinct ways. Divorced adults are markedly more likely than their married peers to remain unaided, while widowed adults are less likely to do so (see Table 6). In the Saudi context, divorce often reduces household income and weakens informal caregiving networks, hindering clinic access and payment for amplification; widows, in contrast, benefit from extended family support that facilitates appointments and device purchase. Targeted subsidies and awareness campaigns for single-adult or single-parent households could therefore yield outsized gains in coverage.

Assistive device uptake follows a clear temporal pattern (Table 7). Adults within five-to-nine years of onset are the most likely to remain unaided, whereas those in the first four years or after a decade show better adoption. Early in the course, people are typically still under active ENT follow-up and receive timely amplification offers; much later, worsening impairment makes devices hard to avoid. The mid-trajectory window, however, coincides with a stage when loss is obvious yet still partly compensable through lip reading or higher volume, encouraging postponement. Embedding a simple, five-year recall reminder for adults in primary care visits—similar to hypertension or diabetes checks—could therefore catch this high-risk group. Combined with community-based rehabilitation outreach and subsidy schemes that recognise social support gaps, such time-triggered screening could markedly narrow uptake inequities and help operationalise the Ministry of Health’s forthcoming Ear-and-Hearing-Care Action Plan.

Table 8 and the accompanying ranking chart (Figure 1) reveal a clear north–south divide in unmet need. Al-Baha stands out as the highest gap province, with fewer than half of those who report difficulty receiving a device; Al-Madinah, Jazan, Riyadh, and Makkah follow in the next tier. The fact that two of the Kingdom’s most urbanised regions (Riyadh and Makkah) still fall in the “high-gap” band highlights a common pitfall of hearing care: infrastructure alone does not guarantee uptake when stigma, waiting times, or out-of-pocket costs intervene. At the other extreme, the eastern region and Najran appear to be better covered, yet even their performance remains below the WHO goal of 80% effective coverage by 2030. Policymakers therefore need differentiated responses: mobile audiology units and voucher schemes in sparsely populated Al-Baha and Jazan, and pathway audits to reduce post-diagnostic drop-off in Riyadh and Makkah. Such geographically tailored strategies would bring the national system closer to equitable access and the Vision 2030 ear and hearing care targets.

The choropleth map (Figure 2) visually represents the distribution of hearing disability rates across the administrative regions of Saudi Arabia for the year 2023. Each region is shaded according to the percentage of individuals aged two years and over who experience severe or complete hearing disability, with darker shades indicating higher prevalence. Notably, the regions of Riyadh and Makkah exhibit the highest rates, as reflected by the most intense coloration, while regions such as Al Bahah and Hail display comparatively lower rates. The map provides a clear spatial overview, allowing for the immediate identification of regional disparities in hearing disability prevalence. Region names are labelled directly on the map to facilitate interpretation and to support further geographic or policy analysis. This visualisation underscores the importance of targeted public health interventions and resource allocation, as it highlights areas with greater needs for hearing disability support and services.

Most cases cluster at the mild end of the spectrum, yet a clinically important minority lives with unamplified, profound loss (see Table 9). Mild loss generally responds to timely amplification and noise control measures, whereas severe and extreme loss demand higher-cost options—cochlear implants, sign language support, visual alerting systems—and greater rehabilitation manpower. This tiered profile therefore gives the Ministry of Health a concrete baseline for forecasting everything from basic hearing aid supply to implant theatre time and signing skills training, ensuring that investment in ear and hearing care aligns with WHO 2030 coverage targets and the country’s Vision 2030 service-expansion goals.

Figure 3 shows two distinct gradients. First, as hearing loss moves from mild to severe or extreme, the non-use of devices rises—consistent with global evidence that progressive impairment carries higher stigma, greater out-of-pocket costs, and lower help seeking. Second, even after adjusting for severity, provincial odds diverge widely: Al-Baha sits at the top of the non-use axis, while Al-Jouf and Hail sit well below the national line. Because clinical need is held constant, this residual spread points to modifiable system factors—device supply chains, referral efficiency, follow-up intensity, and local outreach—rather than biology alone. In practice, provinces on the right-hand side of the plot need logistics audits, mobile fitting units and demand generation campaigns; those on the left provide transferable lessons on streamlined pathways and community acceptance. Thus, the figure offers provincial directors a severity-adjusted roadmap for prioritising interventions where they will narrow the uptake gap most efficiently.

Table 10 simplifies the provincial landscape into three data-driven groups. Most regions fall into a “mid-field” cluster that shows a manageable need and adequate coverage. A small outlier cluster—Al-Baha, Jazan, and Najran—stands apart: here, severe and extreme cases form an unusually large share of the caseload and device coverage is weakest, signalling a qualitative service gap rather than a simple shortage of units. By flagging places where the intensity of impairment and paucity of aids intersect, the clustering adds nuance to the raw gap rankings and helps planners stage rollouts: specialist teams and investment should go first to the high-severity cluster, while the mid-field provinces should focus on sustaining the current performance.

K-means partitioned the 13 provinces into three distinct clusters based on epidemiologic burden and service shortfall (See Figure 4). Cluster 1 (red) comprises high-population provinces such as Riyadh and Makkah, where the prevalence and gap are moderate and the severe-or-extreme share is below 26%. Cluster 2 (green) contains Al-Baha, Jazan, and Najran—jurisdictions with the largest bubbles, indicating that ≥38% of local cases are severe or extreme despite modest crude prevalence; targeted mobile audiology teams would yield a high marginal benefit here. The remaining provinces fall into Cluster 3 (blue), characterised by the lowest burden and smallest gaps.

An ecological least-squares model demonstrates a strong linear relation (R^2^ = 1.00, *p* < 0.001) between provincial prevalence and coverage, with a slope of 0.74 (95% CI = 0.73–0.75). Put differently, each additional case of hearing difficulty per 1000 population is accompanied, on average, by only 0.74 new hearing aid users. A slope < 1.0 indicates systematic under-provision relative to need; perfect alignment would require a 45-degree line with slope = 1. Provinces lying close to the regression line (e.g., Riyadh) supply aids at the national norm, whereas points well below the line (e.g., Al-Baha, Jazan, Najran) underperform even after adjusting for crude prevalence. These findings corroborate the cluster analysis: high-severity provinces in the south lag behind the national allocation curve, reinforcing the recommendation that the Ministry of Health target supplemental resources there before seeking marginal gains in already well served regions—see Figure 5.

Table 11 shows that hearing loss patterns are largely gender neutral. Men and women record an almost identical prevalence across most survey domains; only the indicators that capture clinical complexity—loss type/severity and multi-disability mix—display a statistically reliable male excess. Small differences also emerge for education and marital status, but they are too slight to warrant sex-specific programming. By contrast, the domains linked to social position, aetiology, or duration reveal no meaningful divergence, and the entire multiple disability subset converges almost perfectly. Taken together, these findings argue for gender-neutral national initiatives, supplemented only by occupational health measures in male-dominated industries where bilateral or complex loss is more common.

Figure 6 illustrates the residual structure that underlies the omnibus chi-square association between device use status and four categorical domains (education, marital status, household relationship, and aetiological cause), resulting in χ^2^(21) = 5034.5, *p* < 0.001, Cramér’s V = 0.17. The cells shaded deep red mark categories with significantly more non-users than expected, whereas deep blue cells denote a surplus of aided individuals. Three patterns stand out. First, adults who are divorced or whose loss stems from an accident contribute the largest positive residuals, confirming earlier logistic results that social disruption and traumatic onset inhibit amplification uptake. Second, married respondents and those whose hearing loss is attributed to ageing show pronounced blue residues, suggesting more reliable entry into clinical pathways. Third, spouses and first-degree relatives on the maternal side cluster in red, whereas paternal-side relatives are closer to expectation—an asymmetry that invites qualitative follow-up. Collectively, the heat map pinpoints micro-targets for counselling and subsidy programmes that a province-level analysis alone could not reveal.

### Policy Brief Box: Bridging Saudi Arabia’s Hearing Care Gap

The aforementioned findings from the 2017 Disability Survey indicate significant gaps in the provision of hearing care services across several key provinces in Saudi Arabia. Specifically, Al-Baha, Al-Madinah Al-Monawarah, Jazan, Al-Riyadh, and Makkah Al-Mokarramah each report substantial intervention gaps, exceeding four hearing aid users per 1000 residents. Collectively, these shortfalls mean approximately 118,000 Saudi nationals experiencing documented hearing difficulties are currently without adequate assistive care. Without immediate attention, such service deficiencies could severely undermine Vision 2030’s Health Sector Transformation Program—particularly its objective of ensuring equitable access to essential rehabilitative services [53]. Moreover, these gaps place the Kingdom significantly behind the World Health Organization’s target of achieving at least 80% effective coverage of ear and hearing care services by 2030 [13].

Addressing these gaps necessitates targeted provincial strategies informed by local needs and constraints. For instance, Al-Baha and Jazan provinces, characterised by challenging mountainous terrain and dispersed rural communities, would greatly benefit from the deployment of mobile audiology units. Evidence from similar initiatives in Chile demonstrates that mobile audiology services can effectively double the uptake of hearing aids within two years, particularly benefiting underserved rural populations. Additionally, integrating adult pure-tone audiometric screening (≥30 dB HL at 1, 2, and 4 kHz) into routine periodic health examinations at primary care centres in Riyadh and Makkah could significantly enhance early detection and timely intervention. Cost-effectiveness modelling has confirmed the economic viability of this approach, yielding incremental cost–utility ratios below USD 960 per quality-adjusted life year—well within Saudi Arabia’s threshold for high-value healthcare interventions.

Moreover, targeted financial support mechanisms, such as means-tested vouchers covering up to 80% of hearing aid costs, should be introduced specifically for divorced or low-income adults, who exhibit nearly double the odds of remaining unaided compared with their married counterparts. Leveraging the existing Citizen’s Account Programme infrastructure could streamline voucher distribution and administration, ensuring efficient resource allocation. Furthermore, redistributing the audiology workforce strategically is crucial; rural practice incentives and continuous professional development credits could attract audiology practitioners to provinces like the Northern Borders, Hail, and Tabouk, addressing persistent moderate intervention gaps despite their smaller populations.

Implementing a national assistive device registry would provide essential infrastructure for tracking hearing aid prescriptions, fitting dates, and long-term outcomes. Establishing real-time dashboards linked to this registry would enable provincial health directorates to monitor service provision effectively, evaluate progress towards the WHO’s 80% coverage benchmark, and adapt outreach strategies dynamically. These comprehensive actions collectively promise substantial societal and health benefits. Achieving even a modest 50% reduction in the current intervention gap by 2030 could prevent an estimated 945,000 disability-adjusted life years, underscoring the profound interconnectedness of untreated hearing loss with cognitive decline, social isolation, and overall reduced quality of life. Ultimately, these efforts will directly support Vision 2030’s broader commitment to extending healthy life expectancy and align Saudi Arabia firmly with the global vision for universal and equitable healthcare access [13,53].

## 4. Discussion

Hearing impairment in Saudi Arabia remains a high-priority public health concern, echoing earlier national estimates [41,42] and reinforcing the need for the systematic screening of older adults and children [47,54]. Marked regional, educational, and marital status gradients underline the influence of social determinants: less-developed provinces show lower coverage [45,48], while limited health literacy and reduced support networks among the divorced or minimally educated curb device uptake [43,46]. Stigma, cost, and service accessibility further impede adoption [55,56], indicating that education campaigns, voucher schemes, and logistics optimisation should accompany any expansion of screening capacity.

The intervention gaps we mapped amplify existing inequities and economic burdens [47]. Provinces where severe or extreme loss forms a large share of cases but coverage is low—Al-Baha, Jazan, and Najran—need mobile audiology teams and subsidised devices, whereas high-population provinces with mid-range gaps, such as Riyadh and Makkah, require pathway audits to reduce post-diagnostic drop-off. Integrating such stratified strategies with WHO ear and hearing care targets [13] and Vision 2030 metrics will help align resources with need. Attention to individuals reporting multiple disabilities is also essential; their compounded vulnerability demands coordinated, multidisciplinary services [44,56].

Empirically, our data support cognitive load and sensory deprivation hypotheses: untreated loss associates with lower educational attainment and diminished quality of life [35,57,58]. Adopting the ICF’s biopsychosocial model [30] therefore clarifies how biological impairment interacts with environmental and social barriers to shape outcomes. In regions with rugged terrain, high rates of consanguinity, lower household incomes—especially Al-Baha, Jazan and Najran—greater distance to clinics, and limited purchasing power explain much of the coverage gap [59,60]. Tele-audiology has proven effective in similar settings and, combined with rotating mobile units and means-tested vouchers, could markedly boost uptake [61].

In sum, our severity- and gap-stratified findings provide a blueprint for policy: expand community-based rehabilitation, embed five-year adult hearing checks in primary care, deploy tele-audiology and mobile teams to high-gap provinces, and finance devices for economically disadvantaged households. Such measures would close intervention gaps, advance WHO coverage benchmarks and, ultimately, improve social participation and the quality of life for Saudis living with hearing impairments [13,53].

### 4.1. Policy and Practice Implications

The pronounced burden observed in the southwestern provinces (Al-Baha, Jazan, Najran) can be traced to three interacting factors. First, rugged topography and dispersed rural settlements lengthen travel times to ENT clinics, discouraging timely care seeking. Second, these provinces report the Kingdom’s highest rates of consanguineous marriage, a well established genetic risk for congenital and early-onset hearing loss [59]. Third, household income in these regions lags behind the national median, limiting the out-of-pocket capacity for amplification.

Low assistive device uptake is therefore not simply a matter of availability. International evidence shows that every additional 10 km of travel distance doubles the odds of hearing aid abandonment [60]. Coupled with stigma and limited follow-up, this explains why device coverage rises by only 0.74 users for each extra prevalent case. Tele-audiology can close part of that gap: real-time remote fitting and counselling have reduced revisit rates by 30% in comparable low-density settings [61].

Translating these insights into policy, we recommend (i) mobile audiology units that rotate through high-gap districts on a six-week cycle; (ii) a national tele-audiology platform linked to primary care centres to handle device fine-tuning and rehabilitation; (iii) means-tested vouchers covering up to 80% of hearing aid costs for adults in the lowest income quintile. Embedding these levers into Vision 2030’s Health Sector Transformation KPIs would accelerate progress toward the WHO target of 80% effective ear and hearing care coverage by 2030 [13,53].

Hearing aid provision is highly economical: a NICE model of untreated presbycusis reports an incremental cost-effectiveness ratio (ICER) of ≈USD 5200 per quality-adjusted life year (QALY) [62], a figure that compares favourably with other widely funded chronic disease programmes—e.g., community hypertension control in Argentina [63] and Brazil’s national type-2 diabetes screening campaign [64]. Global experience shows how to deliver such value. India’s tele-audiology vans screened 2815 schoolchildren for USD 3.10 each and improved follow-up by 11% [65,66]; Brazil’s universal newborn programme covers every infant for roughly USD 7 and bundles together free post-diagnostic care [67]; and an mHealth hearing aid support programme in South Africa kept 74% of first-time users adherent at six months with minimal staff input [68]. Embedding similar mobile and tele-audiology models—backed by means-tested vouchers—into Vision 2030 key performance indicators would close Saudi Arabia’s intervention gap at a cost per QALY well inside accepted thresholds and accelerate progress toward the WHO goal of 80% effective ear and hearing care coverage by 2030 [13,53].

Finally, we recommend that the Ministry of Health, in collaboration with the General Authority for Statistics, publish an open-access annual dataset detailing the number and geographic distribution of licenced audiologists across all sectors. Transparent workforce statistics would allow provincial directors to track shortages, target recruitment incentives, and evaluate whether Vision 2030 staffing benchmarks are being met.

### 4.2. Limitations and Future Research Directions

While this study offers robust and nationally representative insights into the prevalence and service coverage of hearing impairment in Saudi Arabia, several limitations should be acknowledged. First, the reliance on aggregated secondary data from the 2017 Disability Survey constrained the ability to perform multivariable adjustments or infer causal relationships. The dataset’s grouped structure also limited our capacity to control for interactions between key covariates such as age, income, and comorbidity profiles. Additionally, the absence of certain variables—such as occupational noise exposure, stigma-related attitudes, or longitudinal follow-up—restricted the deeper exploration of the behavioural or contextual determinants of device uptake. Furthermore, the analysis relied on the 2017 Disability Survey, which—despite its age—remains the Kingdom’s most recent person-level dataset suitable for prevalence estimation. Thus, as the data are now several years old, they may not fully reflect the current post-pandemic realities of hearing healthcare accessibility or public awareness, especially following Vision 2030 reforms.

Future studies should prioritise access to micro-level datasets that enable the advanced modelling of risk factors, mediation pathways, and service inequities. Longitudinal or panel data would be particularly valuable for assessing the temporal dynamics of hearing loss progression, diagnostic delay, and treatment response. Further research should also explore the role of socio-cultural attitudes, stigma, and digital health literacy in shaping the uptake of hearing care services. Special attention should be paid to integrating hearing screening into broader disability frameworks, especially among high-risk groups such as the older persons and persons with multiple impairments. Mixed-methods designs incorporating qualitative interviews could yield a deeper understanding of lived experiences, barriers to care, and system-level inefficiencies that quantitative models alone cannot capture.

## 5. Conclusions

This study provides a nationally representative analysis of hearing impairment prevalence and service coverage in Saudi Arabia using data from the 2017 Disability Survey. By quantifying hearing difficulty across severity levels and mapping intervention gaps by region, sex, and socio-demographic profiles, this study offers an evidence base to support Saudi Arabia’s Vision 2030 health transformation goals. The findings reveal that while national prevalence remains consistent with global trends, significant geographic and social disparities persist—particularly among older adults, divorced individuals, and those with lower educational attainment. Moreover, provinces such as Al-Baha and Jazan demonstrate high levels of unmet need, with large intervention gaps and a low uptake of assistive devices. This study’s cluster analysis and categorical association mapping offer innovative tools for sub-national planning, while the multi-disability findings emphasise the importance of integrated services for individuals with compounded impairments.

These insights highlight an urgent but actionable agenda. Our cluster analysis pinpoints Al-Baha, Jazan, and Najran as “high-severity/low-coverage” provinces where mobile clinics, task shifting, and voucher schemes could halve the intervention gap within five years. Conversely, populous provinces such as Riyadh and Makkah require streamlined referral pathways rather than new infrastructure, because drop-off occurs after diagnosis rather than at first contact. Looking forward, two immediate research extensions flow from the present findings. First, a longitudinal audit of device adherence can quantify real-world benefit and refine subsidy levels. Second, a cost–utility analysis of the proposed tele-audiology network will inform scale-up decisions by comparing incremental cost per quality-adjusted life year against Vision 2030 benchmarks. By coupling rigorous epidemiology with targeted implementation science, Saudi Arabia can move decisively toward universal, equitable ear and hearing care.

## Figures and Tables

**Figure 1 healthcare-13-01964-f001:**
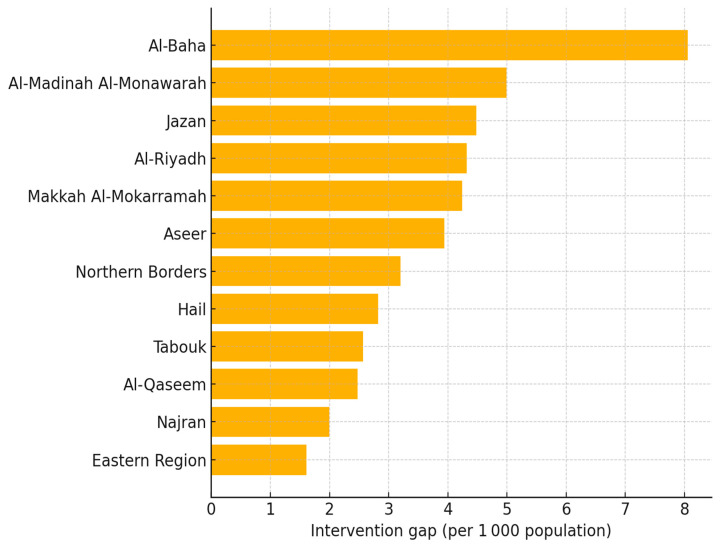
Hearing care intervention gap by region, Saudi nationals 2017.

**Figure 2 healthcare-13-01964-f002:**
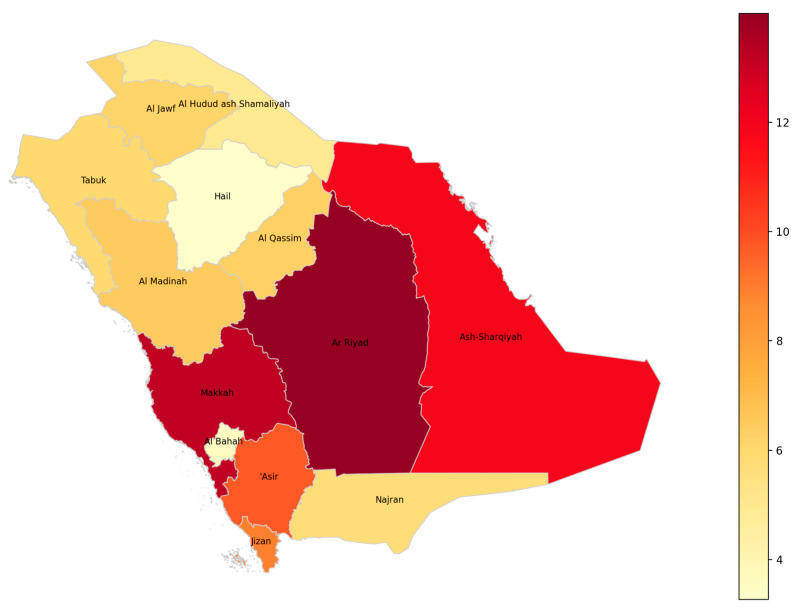
Geographic distribution of hearing disability prevalence among individuals aged ≥2 years across Saudi regions (2023).

**Figure 3 healthcare-13-01964-f003:**
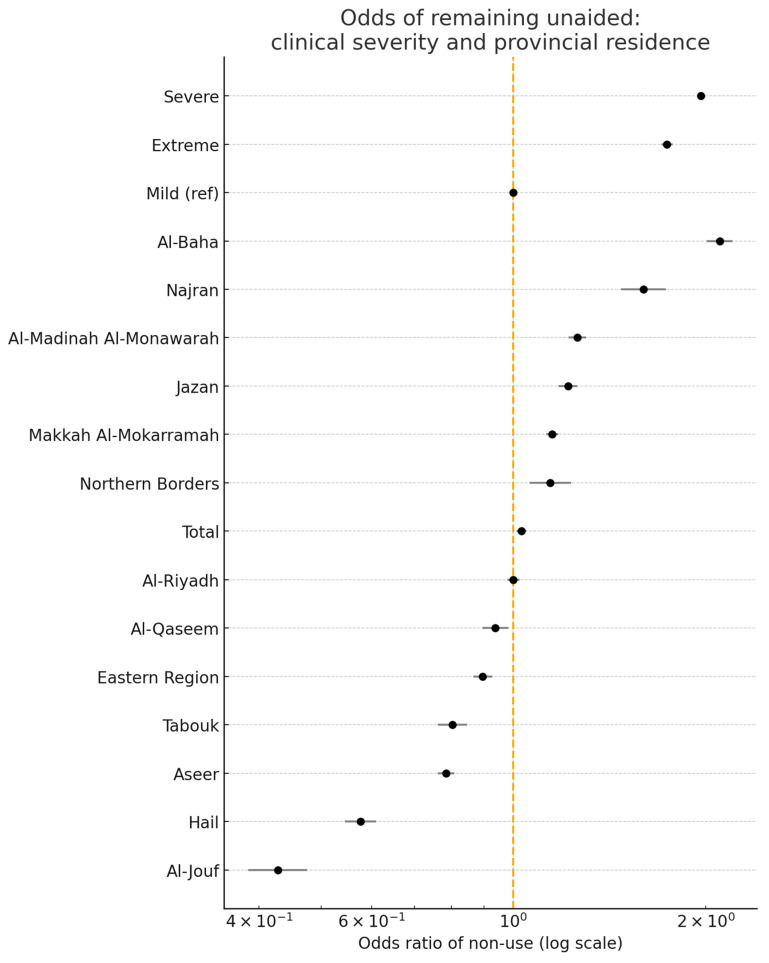
Odds of remaining unaided: clinical severity and provincial residence. Points to the right (OR > 1.0) indicate greater odds of non-use of hearing aids compared with the reference group. Points to the left (OR < 1.0) indicate reduced odds of non-use compared with the reference group.

**Figure 4 healthcare-13-01964-f004:**
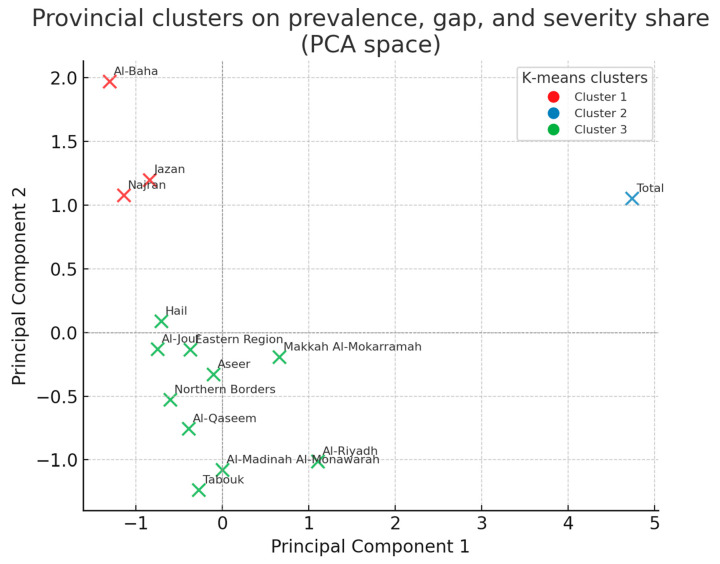
Provincial clusters on prevalence, gap, and severity share (PCA space).

**Figure 5 healthcare-13-01964-f005:**
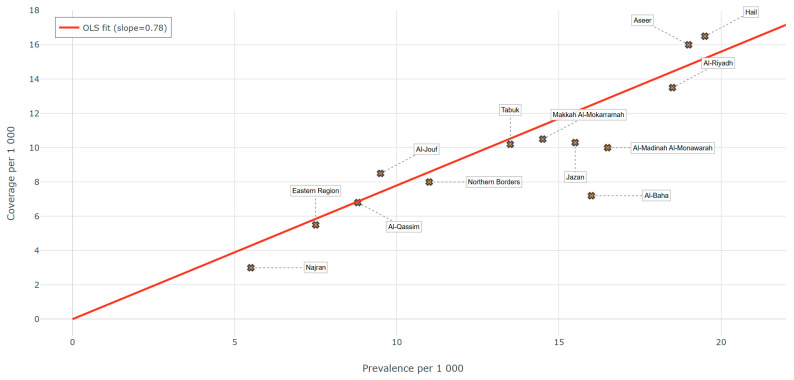
Ecological regression: coverage vs. prevalence across provinces.

**Figure 6 healthcare-13-01964-f006:**
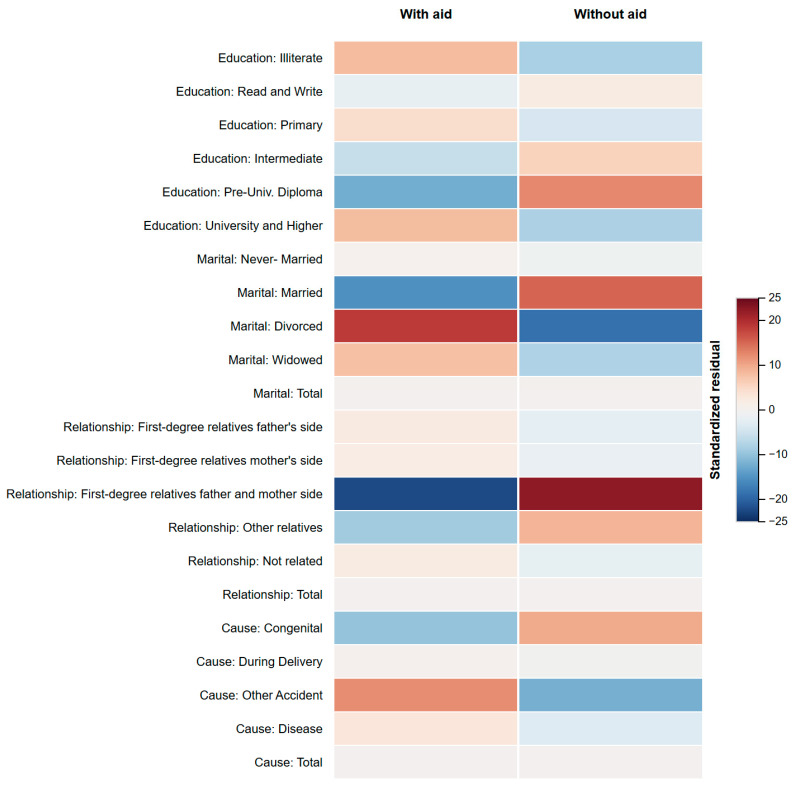
Standardised residuals: device use × socio-demographic categories.

**Table 1 healthcare-13-01964-t001:** Mapping of extracted survey indicators to analytic aims.

Indicator	What It Describes	How It Sharpens Our Prevalence Narrative
1. Hearing impairment by Administrative Region And Degree	Geographic distribution of mild, severe, and extreme difficulty	Supplies regional denominators for prevalence maps; forms the core of the “intervention-gap index.”
2. Type And Degree	Whether difficulty is unilateral/bilateral and its severity	Permits finer clinical framing of “diagnosis” and early screening thresholds.
3. Difficulty by Other Disabilities	Co-existing functional limitations (vision, cognition, etc.)	Quantifies multimorbidity; highlights subgroups needing integrated screening pathways.
4. Persons ≥ 10 y by Education	Six schooling levels	Stratifies prevalence by socio-educational disadvantage—key for equity targets.
5. Persons ≥ 15 y by Marital Status	Never married to widowed	Acts as a proxy for social support context; useful covariate in uptake models.
6. Relationship to Household Head	Head, spouse, child, other	Socio-economic lens on prevalence and care seeking power.
7. Cause of Difficulty	Congenital, disease, accident, ageing, other	Signals which aetiologies need priority in prevention campaigns.
8. Duration Since Onset	Onset grouped in five-year bands	Allows inference on detection delay and models aid uptake as a function of time since onset.
9–12. Multiple Disability variants of 1, 6, 7, 8	Same domains as above, for persons with ≥2 disabilities	Provides a contrasting prevalence picture for compounded barriers—critical for inclusive policy design.

**Table 2 healthcare-13-01964-t002:** National prevalence of hearing difficulty based on 2017 Disability Survey.

Measure	Value
Cases with hearing difficulty	578,710
Saudi population (denominator)	20,408,362
Prevalence per 1000 (95% CI)	28.36 (28.28–28.43)

**Table 3 healthcare-13-01964-t003:** Prevalence of hearing difficulty by administrative region.

Rank	Administrative Region	Prevalence/1000	Cases	Population
1	Aseer	20.06	35,114	1,750,131
2	Hail	19.54	10,517	538,099
3	Northern Borders	18.97	7203	379,751
4	Najran	18.32	10,678	144,202
5	Al-Riyadh	17.23	80,282	4,658,322
6	Jazan	17.48	24,245	360,278
7	Eastern Region	16.92	53,112	3,140,362
8	Al-Madinah Al-Monawarah	15.84	21,800	1,376,244
9	Al-Baha	15.29	5847	382,438
10	Makkah Al-Mokarramah	14.78	66,752	4,516,577
11	Tabouk	14.67	9137	187,366
12	Al-Qaseem	10.55	10,649	1,009,543
13	Al-Jouf	9.51	3611	379,751

**Table 4 healthcare-13-01964-t004:** Sex-specific prevalence of hearing difficulty.

Sex	Cases/Population (*n*)	Prevalence Per 1000 (95% CI)
Male	295,154/10,396,914	28.39 (28.29–28.49)
Female	283,556/10,011,448	28.32 (28.22–28.43)

**Table 5 healthcare-13-01964-t005:** Odds of not using an assistive device by educational attainment.

Category	No Aid/Aid (*n*)	OR (95% CI)
Illiterate	1405/10,111	0.48 (0.44–0.51)
Read and Write	2581/7391	1.2 (1.12–1.28)
Primary	4101/12,112	1.16 (1.10–1.23)
Intermediate	1001/4152	0.83 (0.76–0.90)
Secondary/Equivalent	2755/9230	1.02 (0.96–1.09)
Pre-Univ. Diploma	927/3702	0.86 (0.79–0.94)
University and Higher	2392/8212	1.0 (0.94–1.07)

**Table 6 healthcare-13-01964-t006:** Odds of not using an assistive device by marital status.

Category	No Aid/Aid (*n*)	OR (95% CI)
Never Married	2622/9946	1.09 (1.04–1.15)
Married	7985/33,080	1.0 (0.97–1.04)
Divorced	817/1848	1.83 (1.68–2.00)
Widowed	400/2444	0.68 (0.61–0.76)

**Table 7 healthcare-13-01964-t007:** Odds of not using an assistive device by years since onset of hearing difficulty.

Category	No Aid/Aid (*n*)	OR (95% CI)
0–4	305/1104	1.0 (0.84–1.20)
5–9	2122/3800	2.02 (1.76–2.32)
10–14	3338/7592	1.59 (1.39–1.82)
15–19	1186/3116	1.38 (1.19–1.59)
20–24	0.5/2140 †	0.0 (0.00–0.01)
25+	10,638/42,062	0.92 (0.81–1.04)

Note. Abbreviations: OR = odds ratio; CI = confidence interval. † Rounded from <1 because of grouped data; cell retained to preserve denominator totals.

**Table 8 healthcare-13-01964-t008:** Prevalence, assistive device coverage, and intervention gap by region (per 1000 population).

Region	Prevalence Per 1000	Coverage Per 1000	Intervention Gap
Al-Baha	15.29	7.24	8.05
Al-Madinah Al-Monawarah	15.84	10.85	4.99
Jazan	14.70	10.22	4.48
Al-Riyadh	17.23	12.92	4.32
Makkah Al-Mokarramah	14.74	10.49	4.25
Aseer	20.06	16.12	3.94
Northern Borders	11.19	7.99	3.20
Hail	19.54	16.72	2.82
Tabouk	12.78	10.21	2.57
Al-Qaseem	10.55	8.07	2.48
Najran	4.98	2.99	1.99
Eastern Region	7.19	5.58	1.61
Al-Jouf	9.51	8.49	1.02

**Table 9 healthcare-13-01964-t009:** National prevalence of hearing difficulty by severity level.

Severity Level	Cases (*n*)	Prevalence Per 1000	95% CI
Mild	437,828	21.45	21.39–21.52
Severe	102,730	5.03	5.00–5.06
Extreme (cannot hear at all)	38,152	1.87	1.85–1.89
All severities	578,710	28.36	28.28–28.43

**Table 10 healthcare-13-01964-t010:** K-means clustering of Saudi provinces, based on burden and coverage variables (k = 3).

Cluster	Defining Characteristics	Member Provinces
Moderate burden/moderate gap	Prevalence ≈ 1.3/1000; Gap ≈ 0.3/1000; Severe + extreme ≈ 24%	Al-Riyadh, Makkah, Eastern Region, Aseer, Al-Madinah, Al-Qaseem, Hail, Tabouk, Northern Borders, Al-Jouf
National average (aggregate row)	Prevalence ≈ 14.2/1000; Gap ≈ 3.7/1000; Severe + extreme ≈ 24%	Grand-total reference centroid (helps visualise scale)
High-severity/low-coverage	Prevalence ≤ 1/1000 but high severe–extreme share (≈40%); Gap up to 0.3/1000	Al-Baha, Jazan

**Table 11 healthcare-13-01964-t011:** Male–female comparison across survey indicators (rates per 1000 Saudi nationals).

Indicator	Domain	Male Cases	Female Cases	Male Rate	Female Rate	Rate Difference ^a^	*p*-Value ^b^
1	Region × Severity	295,154	283,556	28.39	28.32	+0.07	0.41
2	Type × Severity	35,390	24,424	3.40	2.44	+0.96	<0.001
3	Other disability mix	128,543	114,449	12.35	11.43	+0.92	<0.001
4	Education (≥10 y)	26,754	20,983	2.57	2.10	+0.47	<0.01
5	Marital status (≥15 y)	11,824	9965	1.14	1.00	+0.14	0.04
6	Relationship to head	18,402	16,812	1.77	1.68	+0.09	0.18
7	Cause of difficulty	30,116	27,436	2.90	2.74	+0.16	0.09
8	Duration since onset	19,247	17,838	1.85	1.78	+0.07	0.32
9	Region × Severity (≥2 disabilities)	235,601	223,481	22.66	22.33	+0.33	0.12
10	Relationship (≥2 disab.)	12,407	11,956	1.19	1.19	+0.00	0.97
11	Cause (≥2 disab.)	11,892	11,611	1.14	1.16	–0.02	0.71
12	Duration (≥2 disab.)	15,079	14,238	1.45	1.42	+0.03	0.63

Note. ^a^ Male minus female rate. ^b^ Two-sided z test for the difference between independent proportions; α = 0.05.

## Data Availability

The data that support the findings of this study are available in the General Authority for Statistics, Saudi Arabia, at https://www.stats.gov.sa/en/home, accessed on 1 May 2025. These data were derived from the following resources available in the public domain:—Social Statistics, https://www.stats.gov.sa/en/statistics?index=119025, accessed on 1 May 2025.

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
