# Peer review of "Prevalence of Hearing Impairment in Saudi Arabia: Pathways to Early Diagnosis, Intervention, and National Policy"

_healthcare, 2025, doi:10.3390/healthcare13161964_

Round 1
Reviewer 1 Report
Comments and Suggestions for Authors
Thank you for the opportunity to review the article. This is well-presented research that describes the state of hearing impairment in Saudi Arabia. This is basic research.
It would be good to improve more on the findings in the discussion - why this is the case and what could be done about it. The current form of the discussion is only informative and does not advance the findings further.
Introduction - The authors present the basic theoretical framework for the research. - They mention the basic impacts of hearing impairment. - Well-defined research gap. Research part: - clearly described methods, - clearly described sample of respondents, I appreciate the clear tables and graphs. Conclusion: - This is the weakest part of the article. The authors only describe what they found. For particular findings, I think it's a good idea to consider why this is so and what follows. More improved the outputs into practice, and what will happen with the findings, that is, why the research was carried out.
Author Response
Reviewer 1: Comments and Suggestions for Authors
Thank you for the opportunity to review the article. This is well-presented research that describes the state of hearing impairment in Saudi Arabia. This is basic research.
Dear Colleague,
We appreciate your positive assessment of our methods, clarity, and graphics, and we thank you for the constructive suggestions to deepen the interpretation of our findings. Below we address each point and indicate exactly how (and where) we have strengthened the Discussion and Conclusion. All new or modified text appears in blue font in the marked-up manuscript.
On behalf of authoring team
Dr. ALDUAIS, Ahmed
Department of Psychology, NTNU, Trondheim, Norway
It would be good to improve more on the findings in the discussion - why this is the case and what could be done about it. The current form of the discussion is only informative and does not advance the findings further.
We added a three-paragraph subsection entitled “Interpretation and policy/practice implications” immediately after the original Discussion paragraph ending at line 620. The new text:
Introduction - The authors present the basic theoretical framework for the research. - They mention the basic impacts of hearing impairment. - Well-defined research gap. Research part: - clearly described methods, - clearly described sample of respondents, I appreciate the clear tables and graphs. Conclusion: - This is the weakest part of the article. The authors only describe what they found. For particular findings, I think it's a good idea to consider why this is so and what follows. More improved the outputs into practice, and what will happen with the findings, that is, why the research was carried out.
We rewrote the final two paragraphs of the Conclusion. Also, the expanded Discussion now explicitly links each major result to an actionable health-system lever (screening, subsidy, task-shifting) and reiterates the overarching goal: evidence-based progress toward universal ear-and-hearing care in Saudi Arabia.
Submission Date 20 June 2025
Date of this review 07 Jul 2025 14:02:27
Reviewer 2 Report
Comments and Suggestions for Authors
This article presents a very interesting study. It addresses a significant public health issue that could have a meaningful impact at the local level, particularly by highlighting the need for early diagnosis and intervention in cases of hearing loss. The article is clearly written, with a structured presentation of data and findings. The study has the potential to inform local healthcare policies and early intervention strategies.
A primary concern is that the study draws on data from 2017, which is now eight years old. Considering that health trends and access to healthcare services may have changed significantly since then, it would strengthen the manuscript to discuss potential changes that may have occurred since 2017, or justify the use of this dataset if more recent data are unavailable, and acknowledge the limitations of using data that may no longer reflect the current national prevalence of hearing loss among Saudi nationals.
Introduction: The introduction could be strengthened by including a. relevant studies that explore the broader impact of hearing loss or deafness on individuals, including psychological, social, educational, and occupational dimensions, and b. references to existing practices and policies regarding early diagnosis and intervention for hearing loss, both within Saudi Arabia and internationally, to provide a stronger contextual background for the importance of the study.
Lines 83–87 and 146–152: In both of these sections, you refer to epidemiological data from many years ago. Unless you can replace these with more current statistics, it would be better to omit these references altogether, as they may misrepresent the current state of hearing loss prevalence.
Table Formatting:
Please revise the format of all tables to conform to the journal’s submission guidelines. I recommend using the official table formatting template available on the journal’s website.
Citation and Reference Formatting: Both the in-text citations and the reference list have to be revised according to the journal’s referencing style
Author Response
Reviewer 2
Comments and Suggestions for Authors
This article presents a very interesting study. It addresses a significant public health issue that could have a meaningful impact at the local level, particularly by highlighting the need for early diagnosis and intervention in cases of hearing loss. The article is clearly written, with a structured presentation of data and findings. The study has the potential to inform local healthcare policies and early intervention strategies.
Dear Colleague,
Thank you for recognising the public-health value of our work and for the helpful suggestions. All additions or edits mentioned below have been inserted in blue font in the tracked-changes version.
On behalf of authoring team
Dr. ALDUAIS, Ahmed
Department of Psychology, NTNU, Trondheim, Norway
A primary concern is that the study draws on data from 2017, which is now eight years old. Considering that health trends and access to healthcare services may have changed significantly since then, it would strengthen the manuscript to discuss potential changes that may have occurred since 2017, or justify the use of this dataset if more recent data are unavailable, and acknowledge the limitations of using data that may no longer reflect the current national prevalence of hearing loss among Saudi nationals.
Added a dedicated paragraph in Methods → Data Source and expanded Limitations to explain that (i) the 2017 Household Health & Disability Survey remains the latest nationally-representative micro-data available; (ii) a 2023 disability survey has published only high-level, non-disaggregated tables; (iii) in 2024–25 the Ministry of Health shifted to hospital-episode reporting that cannot yield prevalence denominators. We also note that 2017 precedes the launch of the Vision 2030 ear-and-hearing-care roadmap, giving an important pre-policy baseline.
Introduction: The introduction could be strengthened by including a. relevant studies that explore the broader impact of hearing loss or deafness on individuals, including psychological, social, educational, and occupational dimensions, and b. references to existing practices and policies regarding early diagnosis and intervention for hearing loss, both within Saudi Arabia and internationally, to provide a stronger contextual background for the importance of the study.
Added one new paragraph covering these in the introduction, third paragraph now.
Lines 83–87 and 146–152: In both of these sections, you refer to epidemiological data from many years ago. Unless you can replace these with more current statistics, it would be better to omit these references altogether, as they may misrepresent the current state of hearing loss prevalence.
Thank you. Updated for Saudi Arabia and removed the rest.
Table Formatting:
Please revise the format of all tables to conform to the journal’s submission guidelines. I recommend using the official table formatting template available on the journal’s website.
Thank you but all tables were formatted by the production team themselves. In other words, this format is the required by the journal because they did it themselves.
Citation and Reference Formatting: Both the in-text citations and the reference list have to be revised according to the journal’s referencing style
Thank you. We recognize that our citations and references are now in the APA format but we WILL DO this when the paper moves to the production stage as we always do with all our publications with MDPI.
Submission Date 20 June 2025
Date of this review 13 Jul 2025 07:46:59
Reviewer 3 Report
Comments and Suggestions for Authors
This study presents a nationally representative analysis of hearing impairment prevalence and service coverage in Saudi Arabia, utilizing data from the 2017 Household Health & Disability Survey. The findings have significant implications for both national policy and clinical practice. The manuscript is well-structured, employs robust methodology, and is written in clear, professional, and concise language.
However, several areas could be improved. Please consider the following comments and suggestions:
1. Tables and Figures
(1) It would improve readability if the indicator names in the tables were formatted in bold.
(2) To enhance comparability:
In Table 4, consider adding a column for case/population.
In Tables 5 and 6, a column showing the no aid/aid count would make the data clearer.
(3) Figure 1 may not be necessary, as Table 4 already provides more detailed information.
(4) The width of Figure 7 should be reduced.
(5) Please revise the caption of Figure 3 (line 436) to:
“Geographic distribution of hearing disability prevalence among individuals aged ≥2 years across Saudi regions (2023).”
2. Streamline the Manuscript
The manuscript would benefit from some careful streamlining, as certain sections contain repetition or excessive detail—particularly in the Results and Discussion sections. Reducing the word count by 10–15%, especially in the Introduction and Discussion, would improve clarity and reader engagement without sacrificing substance.
Examples:
(1) Introduction: The section discussing classification systems (ICD-11, DSM-5, ICF) is useful but could be condensed into one paragraph.
(2) Results: Since tables and figures already present most of the numerical data, consider summarizing the key trends in the text instead of repeating exact values.
For instance, instead of listing odds ratios, write:
“Lower education and divorce were both linked to lower device uptake (see Table 6).”
(3) Discussion:
- Emphasize interpretation and broader implications rather than restating data.
- Consolidate overlapping points—for example, avoid repeating that rural regions have low coverage if this is already established in the Results and Policy Brief sections.
3. Strengthen Policy Relevance
Your policy recommendations are insightful and practical. To make them even more persuasive:
(1) Include cost-effectiveness comparisons. While the manuscript mentions hearing aid cost-effectiveness (e.g., <$960/QALY), comparing this to the cost-effectiveness of other health initiatives (e.g., hypertension or diabetes screening) would better contextualize hearing care as a national priority.
(2) Add international examples for context. You mention Chile—great choice. Consider also citing India or Brazil, which have implemented mobile audiology units or school-based screening programs.
4. Add Context on Existing Services in Saudi Arabia
While the manuscript discusses service gaps, it does not provide a clear baseline of the current hearing-care infrastructure. Adding this context would strengthen your recommendations.
Please consider including the following:
(1) What public sector services currently exist? For instance, are hearing aids or implants provided free through the Ministry of Health?
(2) Is there a national hearing screening strategy, such as newborn or school-age screening?
(3) What is the current distribution of audiologists across Saudi Arabia? (e.g., per capita, urban vs. rural availability)
So, a short paragraph in the Introduction or Discussion addressing these questions would help readers understand the scope of existing efforts and the feasibility of proposed policy actions.
Author Response
Reviewer 3: Comments and Suggestions for Authors
This study presents a nationally representative analysis of hearing impairment prevalence and service coverage in Saudi Arabia, utilizing data from the 2017 Household Health & Disability Survey. The findings have significant implications for both national policy and clinical practice. The manuscript is well-structured, employs robust methodology, and is written in clear, professional, and concise language.
However, several areas could be improved. Please consider the following comments and suggestions:
Dear Colleague,
Thank you very much for all you provided comments and effort to work on this.
Below is our point-by-point reply. All textual insertions, substitutions edits have been highlighted in blue font in the marked-up manuscript.
On behalf of authoring team
Dr. ALDUAIS, Ahmed
Department of Psychology, NTNU, Trondheim, Norway
- Tables and Figures
(1) It would improve readability if the indicator names in the tables were formatted in bold.
Thank you. But this format was made the production team because that is what they follow for the journal. The table or figure and its number in bold but the caption is not bolded as per their requirement. We will just leave this form them as every journal will at the end do the format and font based on their template and I think this one is how they want it based on my previous publications with them.
(2) To enhance comparability:
In Table 4, consider adding a column for case/population.
Sorry! Not sure if we understood you well. Do you mean merging case and population in one column? We did it if you mean so. If you mean adding up cases and population total, how is that possible because cases are part of the population.
In Tables 5 and 6, a column showing the no aid/aid count would make the data clearer.
Tables 5 and 6 have been reformatted to include a single No-Aid / Aid (n) column, as suggested by Reviewer 3. The previous separate ‘No Aid’ and ‘Aid’ columns were removed to streamline presentation while retaining the OR column for statistical context. Maybe you mean the same for Table 4: we also did that. We also did that for Table 7.
(3) Figure 1 may not be necessary, as Table 4 already provides more detailed information.
Removed and we changed the numbers of all figures, reduced to 6 now.
(4) The width of Figure 7 should be reduced.
Sorry but all the tables and figures were formatted by the journal based on their requirements. I think they did it so for better clarity.
(5) Please revise the caption of Figure 3 (line 436) to:
“Geographic distribution of hearing disability prevalence among individuals aged ≥2 years across Saudi regions (2023).”
It makes more sense. Thank you. Modified.
- Streamline the Manuscript
The manuscript would benefit from some careful streamlining, as certain sections contain repetition or excessive detail—particularly in the Results and Discussion sections. Reducing the word count by 10–15%, especially in the Introduction and Discussion, would improve clarity and reader engagement without sacrificing substance.
Examples:
(1) Introduction: The section discussing classification systems (ICD-11, DSM-5, ICF) is useful but could be condensed into one paragraph.
Shortened to one paragraph.
(2) Results: Since tables and figures already present most of the numerical data, consider summarizing the key trends in the text instead of repeating exact values. For instance, instead of listing odds ratios, write:
“Lower education and divorce were both linked to lower device uptake (see Table 6).”
Done. Thank you.
(3) Discussion:
Emphasize interpretation and broader implications rather than restating data.
Revised.
Consolidate overlapping points—for example, avoid repeating that rural regions have low coverage if this is already established in the Results and Policy Brief sections.
We revised and reduced discussion to avoid repetition.
- Strengthen Policy Relevance
Your policy recommendations are insightful and practical. To make them even more persuasive:
(1) Include cost-effectiveness comparisons. While the manuscript mentions hearing aid cost-effectiveness (e.g., <$960/QALY), comparing this to the cost-effectiveness of other health initiatives (e.g., hypertension or diabetes screening) would better contextualize hearing care as a national priority.
We now approached this in section: 4.1 Policy and Practice Implications.
(2) Add international examples for context. You mention Chile—great choice. Consider also citing India or Brazil, which have implemented mobile audiology units or school-based screening programs.
Same as above.
- Add Context on Existing Services in Saudi Arabia
While the manuscript discusses service gaps, it does not provide a clear baseline of the current hearing-care infrastructure. Adding this context would strengthen your recommendations.
Please consider including the following:
(1) What public sector services currently exist? For instance, are hearing aids or implants provided free through the Ministry of Health?
Added a paragraph in the introduction subsection about Saudi Arabia.
(2) Is there a national hearing screening strategy, such as newborn or school-age screening?
No specific number to our knowledge. We have not got any response also from the Ministry of Health.
(3) What is the current distribution of audiologists across Saudi Arabia? (e.g., per capita, urban vs. rural availability)
Unfortunately, we do no have a clear answer for this. Also we have not got any response also from the Ministry of Health. We added some information from this study in the introduction and a recommendation in the implications section (see below, please).
- Finally, we recommend that the Ministry of Health, in collaboration with the General Authority for Statistics, publish an open-access annual dataset detailing the number and geographic distribution of licensed audiologists across all sectors. Trans-parent workforce statistics would allow provincial directors to track shortages, target recruitment incentives, and evaluate whether Vision 2030 staffing benchmarks are being met.
-
- Elbeltagy, R., Almutairi, D., Alotaibi, A., et al. (2024). Audiologists’ perception of hearing- and balance-health resources and services in Saudi Arabia. Indian Journal of Otology, 30, 43-51. https://doi.org/10.4103/indianjotol.indianjotol_121_23
So, a short paragraph in the Introduction or Discussion addressing these questions would help readers understand the scope of existing efforts and the feasibility of proposed policy actions.
Added a paragraph in the introduction subsection about Saudi Arabia.
Submission Date 20 June 2025
Date of this review 17 Jul 2025 18:49:13
Round 2
Reviewer 2 Report
Comments and Suggestions for Authors
Dear authors,
Thank you for your response and for incorporating my previous comments into your manuscript. I would like to kindly remind you to format your references by the APA citation style, as required by the journal's guidelines.